# Colony defence in bumblebees (*Bombus terrestris*)

**Sajedeh Sarlak**[1], **Divya Ramesh**[2], **Ahmad Ashouri**[1], **Seyed Hossein Goldansaz**[1], **Alica Schwarz**[2], **Lena Seitz**[2], **Anja Weidenmüller**[2], **Vlad Demartsev**[2,3,4], **Christoph Kleineidam**[2,3], **Morgane Nouvian**[2,3,5]*

1 Department of Plant Protection, College of Agriculture and Natural Resources, University of Tehran, Karaj, Iran, 2 Department of Biology, University of Konstanz, Konstanz, Germany, 3 Center for the Advanced Study of Collective Behaviour, University of Konstanz, Konstanz, Germany, 4 Department for the Ecology of Animal Societies, Max Planck Institute of Animal Behavior, Konstanz, Germany, 5 Zukunftskolleg, University of Konstanz, Konstanz, Germany

* morgane.nouvian@uni-konstanz.de

## Abstract

Bumblebees are key pollinators of both wild plants and agricultural crops, hence understanding their biology is critical for conservation efforts as well as for managing domesticated colonies. While their foraging and reproductive ecology have received a lot of attention, we know little about another crucial part of their behavioural repertoire: colony defence. In this study, we examine the defensive responses of individuals in whole colonies, maintained in the laboratory, after disturbing them with a mechanical jolt. As a result, we present a detailed ethogram of the defensive behaviour of *Bombus terrestris* in response to mechanical disturbances, as could be induced by vertebrate attacks or handling. In addition to identifying and describing the different responses elicited by this disturbance, we provide information about their temporal sequence, their location and the proportion of bees involved. We also determined whether individual bees produce specific combinations of behaviours, which revealed that a core set of responses is exhibited by all bees, with other behaviours being randomly performed. Finally, we show how colony defence varies depending on disturbance type (mechanical jolt, intruder breath or foreign object). Overall, we demonstrate that colony defence includes measures preparing members for a response, searching for the source of the disturbance, warning intruders to maintain distance, potentially signalling the threat to other colony members and finally recovering from the disturbance. This comprehensive overview provides a valuable starting point to further understand how defensive behaviour is regulated such that bumblebee colonies can survive predator attacks and thrive.

**Data availability statement:** The raw data (tabular outputs from Boris scoring and audio features) is available on Dryad under DOI: 10.5061/dryad.jh9w0vtq8.

**Funding:** This research was supported by the Zukunftskolleg at the University of Konstanz and by a grant from the University of Tehran Research Vice-Chancellor, Ministry of Science, Research and Technology, and Iranian National Science Foundation (project number 97011974). The funders had no role in study design, data collection and analysis, decision to publish, or preparation of the manuscript.

**Competing interests:** The authors have declared that no competing interests exist.

## Introduction

With their teddy bear-like appearance and unfazed behaviour, bumblebees are generally regarded as docile, gentle animals that some people even try to pet. They can, however, become fierce defenders when their nest is in danger. Since their venom can trigger severe allergic reactions, understanding this behaviour is important for both conservation efforts in the wild and for their daily management in greenhouses [1]. Yet, research on this topic has been limited. Bumblebees live in relatively small, often subterraneous colonies that contain their queen, workers, brood, and nectar stores. Ants, wasps, and other insects may occasionally prey on the brood and stores [2–4]. The nests are also attractive to other bees, in particular reproductive parasites (queens or fertile workers from other colonies trying to hijack the colony's resources for their own reproduction, including specialists such as cuckoo bees). In contrast with honeybees, resource theft has been reported but seems infrequent among bumblebee nests [5]. In large colonies, some bees stand guard at the entrance and check incoming bees to prevent these intruders from entering [5,6]. Guards tend to be among the largest bees of the colony and can perform this task for several days [5,7,8]. Non-nestmates are recognized through the cuticular lipids that they carry, which label their colony of origin and signal their fertility status [5,6]. *Bombus terrestris* guard bees are surprisingly permissive to sterile non-nestmates, most often merely antennating them for longer. They also display self-grooming upon encountering them. In contrast, they react to fertile non-nestmates by buzzing, abdominal pumping, and moving their front legs while facing the intruder. Overt aggression is rare and involves darting, head-butting, biting and stinging [6]. Honey-daubing (drenching the intruder in honey to impede its movements) has also been reported in three *Bombus* species [9].

Bumblebee nests also have larger predators. Great tits (*Parus major*) have been observed hunting foragers at the nest entrance [10] while European badgers (*Meles meles*) and skunks (*Mephitis mephitis*) may dig up and consume nests in their entirety [2,3,11–13]. According to a survey, people gardening or building caused a quarter of reported nest mortality in rural environments [3]. Finally mice, voles and shrews often share their burrow system with bumblebees and are thought to prey on the incipient nest while the queen is away, before the emergence of the first workers and hence defenders [2,10]. While it is unclear if bumblebees can fight off sturdy badgers and skunks, their painful sting is likely a good deterrent for other predators. Early warning signs of a vertebrate intruder approaching may include mechanical disturbances to the nest (shocks and vibrations) and elevated $CO_2$ levels, and these stimuli are indeed known to trigger defensive responses in bumblebees [14,15]. The defenders exiting the colony direct their attacks against dark objects in close proximity to the nest, thus showing that visual cues also play a role [16].

The behaviours exhibited by workers during disturbances simulated to mimic those caused by large vertebrates are diverse. The most obvious response is an increase in locomotion, be it walking, running or flying. Running on or around the nest was sometimes termed "patrolling" and was often interpreted as a search for the source

of the disturbance [14,15,17,18]. "Perching" describes bees that stand motionless at the nest entrance or on the edge of the brood clump with their antennae raised, which likely reflects their alerted state [14,17]. Another conspicuous defensive behaviour of bumblebees consists of raising one or multiple legs straight above their body. In most cases, the bee simultaneously tilts her whole body sideways and curls her abdomen such that her stinger is pointed at the perceived threat [5,15,18,19]. Threatened bees may also lie on their back, with their mandibles open and their stinger pointing upward [15,18,20]. On top of these responses and very obvious to anyone who ever bothered them, disturbed bumblebees produce sounds by vibrating their thorax or wings. At least two types of defensive sounds have been recorded within the same colony [15,21]. Unfortunately, limited acoustic analysis of these sounds is available such that it is difficult to identify them across studies.

This survey of the literature on bumblebee defensive responses reveals that most of the information is fragmented, such that there is no clear overview of what the complete behavioural sequence entails. Thus, in this study, we set out to provide a comprehensive ethogram of the defensive behaviour of *B. terrestris* in response to potential vertebrate threats or other mechanical disturbances. We describe the elicited behaviours, their site specificity, the proportion of responders and the frequency and duration of the responses. We also examined if some behaviours are produced in combinations by single individuals, as a correlated pattern or even a sequence. Finally, we investigated how the defensive response varied depending on the type of disturbance (mechanical jolt, human breath or foreign object inside the colony). By highlighting the temporal dynamics and adaptive defensive responses of bees, our results provide background for further in-depth analysis of bumblebee colony defence. Understanding how environmental threats and stressors affect these economically important insects is crucial, as their ability to function as pollinators in wild ecosystems depends on their behavioural and physiological responses. Managing their defensive responses is also important for handling them in greenhouse settings, where they are often the sole pollinators used.

## Materials and methods

All the experiments were conducted at the University of Konstanz, Germany, during the spring and summer months of 2021 and 2022.

### Animals

Four commercially reared bumblebee (*Bombus terrestris*) colonies, ordered from Koppert Biological Systems, were used to investigate the defensive behaviour of bumblebees in response to a mechanical disturbance (Part I). In order to investigate the behaviour of a whole functioning colony, twenty workers of medium size and the queen were selected from each mother colony and individually marked on their thorax with a numbered tag (Opalith Zeichenplättchen Leuchtfarben, Bienen-Voigt & Warnholz, Ellerau, Germany). The test colonies were housed in a cylindrical plexiglass arena (height = 10 cm and radius = 9.5 cm) covered by a transparent ceiling. For the sound analysis, to prevent overlapping acoustic signals, we established separate single-worker setups (11 single bees from the same mother colonies) housed in a smaller arena (height = 5 cm, radius = 7 cm). Some brood (including egg-cups, larvae and at least one pupae) was also transferred to each colony (Fig 1). Finally, twelve microcolonies each comprising five workers, housed within the same smaller arenas, were established to study how the defensive responses could vary depending on stimulus type (Part II). A first set of tests compared intruder breath to mechanical disturbance, using four of these microcolonies. They contained bees selected from two new mother colonies. The second set of tests, comparing foreign object to mechanical disturbance, was done on eight microcolonies derived from yet another two mother colonies (four from each). The worker bees in microcolonies laid and raised their own brood.

In all cases, small perforations were present around the arenas to facilitate air circulation and ventilation within the nest. The arenas' floor was lined with cat litter to reduce moisture levels and keep colonies clean. The colony's daily nutritional requirements were met with *ad libitum* access to pollen and sucrose solution. The bees were kept at 24 ± 2 °C in complete

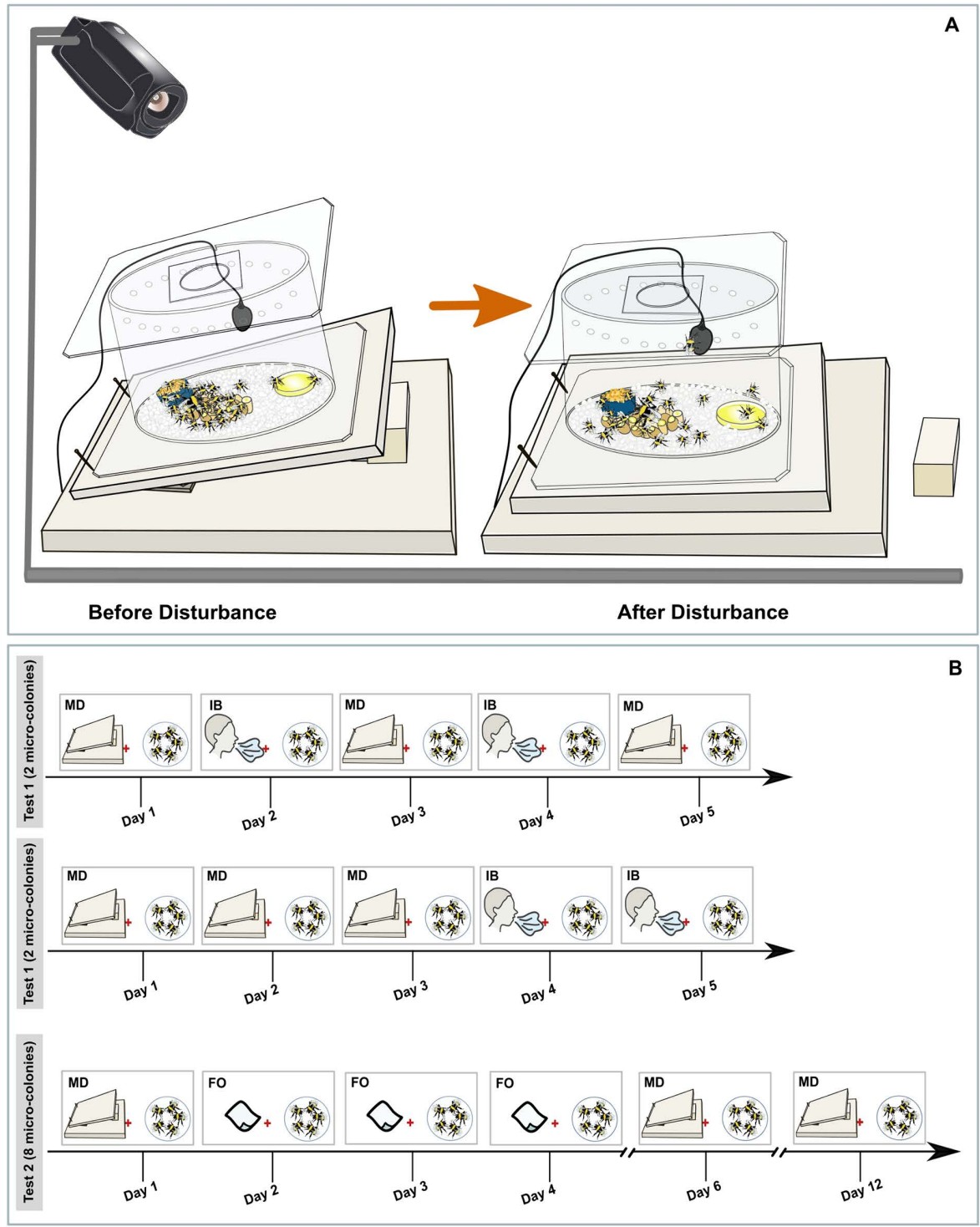

**Fig 1. Methods. A)** The test arena and the mechanical disturbance employed to assess defensive behaviour. Each test colony consisted of 20 workers with a queen and was placed onto the upper wooden board. Before the start of the experiment, the bees were given a 10-minute relaxation period, during which they calmly incubated the brood and exhibited no defensive behaviours. We first observed their behaviour for two minutes while they were

calm, then applied a mechanical disturbance by pulling the wooden cube out. The resulting behaviours were recorded for eight minutes after the disturbance. The entire experiment was recorded using a camera mounted above the testing arena, enabling precise observation of the bees' behaviour in response to the disturbance. **B)** Timeline of Part II, testing the response to different stimuli in microcolonies of five bees. These microcolonies were subjected to a series of stimuli over days, including the mechanical disturbance (MD) and either the intruder breath (IB) or a foreign object (FO) (filter paper).

darkness, except during the experimental manipulation when the lights were turned on. After being moved to the arenas, the bees were given two weeks to acclimate to their new environment before the experiments began. During this time, any newly emerged workers or drones were removed to maintain a constant colony structure.

## Part I: Defensive behaviour in response to a mechanical disturbance in a whole functioning colony

**Disturbing stimulus.** The mechanical disturbance stand consisted of two wooden boards; the smaller board (23 cm × 23 cm) was connected to a larger board (30 cm × 30 cm) by a hinge. Between the two boards, a movable wooden cube with a height of 4 cm was placed, creating an angle of 10° between the upper and lower boards. The experiment was conducted by first placing the housing arena on top of the upper board. To ensure the bees were calm after this movement, we waited for 10 minutes before starting the recording. To disturb the colony, the cube was removed which caused the arena to fall, resulting in a mechanical "clap" (Fig 1A).

**Behavioural analysis.** The behaviour of each colony was observed for 10 minutes, for five consecutive days. Each trial comprised two minutes of observation prior to the disturbance, during which the colonies were calm and in their undisturbed state, and eight minutes of observation following the disturbance (i.e., the mechanical clap). Each trial was recorded using a camera (Canon LEGRIA HF R87) located above the experimental setup (Fig 1A).

Because of time constraints, only the video footage from the first and fifth days of the experiment were analysed to obtain data for this study, thus resulting in 8 analysed trials (4 colonies x 2 days). Video analysis was conducted using the Behavioural Observation Research Interactive Software (BORIS) [22]. Each bee present in the arena was meticulously assessed for nine distinct behavioural states, as per the ethograms detailed in Table 1. A video containing examples of each behaviour is available on Dryad (DOI: 10.5061/dryad.jh9w0vtq8). The location of each bee was also recorded, distinguishing whether they were inside or outside of the nest (defined as the wax structure containing all food stores and

**Table 1. Ethogram of defensive behaviour in *B. terrestris*.**

| Behaviour | Description |
|---|---|
| Abdominal pumping | Fast dorsoventral abdominal movements. |
| Continuous buzzing | Fast and continuous movement of the wings. It usually happens right after the disturbance while the bee is running. |
| Fanning | Steady fanning with spread wings while standing still for at least 10 seconds (removed from most analysis because of very few instances). |
| Flying _ climbing | Flying away or trying to climb over the wall or microphone. |
| Grooming | Grooming of the body by using the mouthparts to cleanse the legs or antennae, and/or the legs to clean the abdomen or wings. |
| Leg raising | The middle and/or hind legs of one side raised or lying on the back with the stinger pointing up. |
| Patrolling | Rapidly walking around or over the nest. |
| Perching | Standing motionless with antennae raised or scanning, in a lookout position. |
| Pulse buzzing | Short and discontinuous movements of the wings while the wings are parallel or with an angle of less than 180° to the axis of the body. |

brood). While the exact identity of the bees could not be maintained across the 2 days of scored recordings due to some of the numbered tags not being readable, each bee was followed individually during a single trial. As a result, we collected a total of 157 behavioural sequences from the 84 bees (80 workers and 4 queens) in our arenas, with most bees contributing 2 sequences. The 11 missing bees were excluded from the analysis because they hid inside/under the nest for some time and hence could not be consistently followed throughout the trial. Moreover, the velocity over time for five random workers (different on each day) and the queen was quantitatively analysed using a bi-dimensional (XY) mode within the Manual Tracking plugin in ImageJ software [23].

**Sound recording and processing.** To collect audio data, we recorded from workers housed individually (see "Animals" section above). We used a microphone (AGPTEK Lavalier Microphone) hanging inside the arena during the test and connected to a computer (HP Pavilion x360) using the inbuilt sound recorder software of Windows 11. The recording was made in stereo with a 48 KHz sampling rate and 32 bits. A total of 11 single bees from the same four mother colonies were recorded for five days. We then down-sampled the files to 22 KHz and 16bit, mono WAV format. For each audio recorded, we generated a spectrogram using Adobe Audition 24.2.0.83 (Adobe Inc., USA) at 512 FFT length, using a Blackman-Harris window. To reduce background noise levels, we used Adobe Audition's built-in noise reduction process feature, by capturing a five-second noise segment and performing a 30 dB spectral noise subtraction on the whole file duration. As an additional step, to further reduce the low-frequency noise, we applied a Butterworth high pass filter of 70 Hz. We inspected the spectrograms and manually labelled all acoustic elements using Adobe Audition markers. We defined the type of the sound according to its spectral and temporal structure and its association with the behavioural state of the individual bee, as scored from the video material.

**Sound annotation verification and acoustic measurements.** As a quality control step, we calculated Signal to Noise Ratio (SNR) for each of the annotated sound segments using the "sig2noise" function from warbleR package [24]. We divided the amplitude of the annotated sound by the mean amplitude of the time segments of equal lengths which occur before and after the annotated sound. The duration of each time segment was equal to that of the annotated sound. SNR = 1 means the signal and background noise have the same amplitude and are difficult to separate spectrally. We selected a conservative threshold of SNR = 1.5 and omitted all sound segments with lower SNR values.

To verify the manual sound type classification, we trained a random forest classification model [25]. We represented the spectral structure of the sound segments using Mel-frequency-cepstral-coefficients (MFCC), which have been successfully applied for animal call classification tasks [26,27]. For each of the sounds (n = 840), we calculated MFCC using the MFCC Function [28]. Each call was segmented into nine equal-length windows with 12 Mel-filters (frequency bins) extracted along each window. The extracted MFCC features were split into a 20 and 80 percent train and test sets respectively. As an internal diagnostic, a reverse prediction of the sound types was performed on the training set. The types of the training set calls were predicted with 100% accuracy, indicating a good predictive power of the random forest model. We then used the trained model to classify the sounds in the test set.

To provide basic acoustic description of the defined sound types we selected six Spectro-temporal metrics: *duration* – length of the sound (sec); *entropy* – energy distribution of the frequency spectrum (pure tone ~ 0; noise ~ 1); *mean frequency* – calculated as the weighted average of the frequency spectrum (kHz). This metric represents the centroid of the frequency distribution, where frequencies with higher energy content exert a greater influence on the average; *dominant frequency* – an average of the frequencies exhibiting the highest energy measured across the spectrogram (kHz). This metric represents the most perceptually salient frequency component of the sound; *F1 frequency* – the frequency for the second harmonic (kHz); *HNR* – harmonic to noise ratio (dB). To extract the spectral features, we used the Spectro-analysis R function from the warbleR package [24].

**Temporal organisation of buzzing bouts.** We measured Inter-Onset Intervals (IOI) from the initiation of a sound element to the initiation of the next one. Since this analysis is less sensitive to the quality of audio recording, we used our full annotated dataset, without omitting calls with low SNR values. To exclude sound events that are less likely to

be part of a continuous behavioural sequence we have calculated a time threshold corresponding to the 95% of IOI distribution (12.8 sec) and omitted from the analysis all IOIs exceeding it. As we were mainly interested in generally characterizing the distinct acoustic profiles of the bee behaviour, we only included symmetric sound pairs in this analysis (both the first and the second sounds are assigned to the same type). We followed previously published procedures for acoustic rhythm analysis [29–31] to calculate the IOI ratios between all pairs of sequential sounds. The calculations were done according to $\frac{I_n}{(I_n + I_{n+1})}$, where "I" is the IOI duration and n is the rank of the IOI in a call sequence. The ratios were categorized according to set small integer rhythm values. For example, when $I_n = I_{n+1}$ the calculated ratio is 0.5 and is assigned to the 1:1 category. When $I_n = \frac{I_{n+1}}{2}$ the calculated ratio is 0.66 ratio and is assigned to the 2:1 category. The category boundaries were set as ± 0.25 from the exact ratio so that ratios between 0.44 (1: 1.25) and 0.55 (1.25: 1) were categorized as on-integer 1: 1 category. Ratios falling on either side of this range 0.4 to 0.44 (1: 1.5 to 1: 1.25) and 0.55 to 0.6 (1.25: 1 to 1.5: 1) were categorized as off-integer 1:1, low and high, respectively. A detailed description and justification of categorical boundaries is available in [31].

To determine whether the observed distribution of IOI ratios deviated significantly from a random distribution, a null distribution was generated by replacing the IOIs in the data with values drawn from a uniform random distribution. To maintain the structure of the data and control for possible individual-level variation, the range of the distribution to be drawn was set separately for each recording, according to its actual minimum and maximum IOI values. Thus, both the null dataset and the data extracted from audio recordings had an identical structure in terms of the number of recorded individuals and sounds. We used the Kolmogorov-Smirnov (KS) test to determine whether the frequency distribution of the IOI ratios was significantly different from the null ratio distribution. All data processing, visualisation and statistical testing were performed in R version 4.1.1 [32].

**Part II: Defensive behaviour in response to different stimuli in microcolonies**

**Disturbance protocol.** This experiment was done on microcolonies of 5 workers, as described in the "Animals" section. Tests were conducted to compare the effects of an intruder's breath, of a foreign object placed inside the bumblebee nest or of the mechanical disturbance. To simulate a disturbance similar to that caused by an intruding mammal, a human's exhaled breath was directed into the arena for approximately five seconds (always by the same experimenter), by slightly lifting the lid of the test arena to blow inside. To mimic a disturbance involving a foreign object inside the nest, a 2.2 cm x 3 cm piece of filter paper was folded and placed near the brood. The mechanical clap described in Part I was used as a reference for comparison with the other types of disturbance.

In the first set of tests, two of the microcolonies received the mechanical disturbance once a day for the first three days of the experiment, and the breath stimulus on the next two days. For the other two microcolonies, the mechanical disturbance and breath stimulus were alternated every other day, for five consecutive days. The second set of tests was conducted over a span of twelve days. All eight microcolonies were exposed to the mechanical clap on day one, day six, and day twelve. On day two, three, and four bees received the foreign object (Fig 1B).

**Behavioural analysis.** The behavioural responses of each worker in the microcolony (see Table 1: flying-climbing, patrolling, continuous buzzing, leg raising, abdominal pumping, grooming, pulse buzzing and its location inside/outside the nest) were observed and recorded for a duration of three minutes. Each disturbance experiment consisted of a minute of observation before the disturbance, during which the microcolonies were in their normal state and calm, followed by two minutes of observation after the disturbance. In the case of the disturbance with a foreign object, the filter paper remained in the nest during the 2 min of disturbance.

**Statistical analysis of behavioural data**

Using the data acquired from the video analysis, the following measures were obtained: 1) Activity level represented by the velocity of tracked bees; 2) Location of the bee with respect to the brood area; 3) Probability of occurrence, duration

of each bout and location (within/outside of brood area) for all scored behaviours. Since the identity of the bee was kept within a single trial for this last type of data, we also examined whether some behaviours are combined into a broader response sequence. We analysed the data in R version 4.4.1 [32]. For Part I, we divided the whole 10 minutes observation period into one minute time bins to enable statistical analysis. Based on observed activity patterns, these bins were then grouped into four broader categories: "before disturbance" (the first two minutes before applying the mechanical clap), "acute response" (the first minute after the disturbance was applied), "delayed response" (the second, third, and fourth minutes following the disturbance), and "recovery" (the fifth, sixth, seventh, and eighth minutes following the disturbance). Since we did shorter recordings for Part II, we had only three time-bins of one min each: "before disturbance", "acute response" and "delayed response".

To describe the changes in bees' response over time for each behaviour, we used generalized linear mixed-effect models (GLMMs; packages: "lme4", "car") with a binomial distribution (when probability was the response variable) or a Gamma distribution and log link function (when the response variable was the duration). Where appropriate, the model included random intercepts for each bee, colony and/or each bee within colonies to account for repeated measures within individuals. The fixed effects were set to be the different time bins. Additionally, the day of the experiment was included as fixed effects whenever it was significant. To analyse the duration that each behaviour was performed, the data was filtered to only include bees that did the behaviour. Because of very few occurrences, a Fisher's exact test was used to examine pairwise differences in the probability of the "leg raising" behaviour across different time bins. The p-values were adjusted for multiple comparisons using the Bonferroni method. A generalized linear mixed-effects model (GLMM) with the time bin and bee type (queen or worker) as fixed effects and the bee identity as random intercept with a Gamma distribution was used to investigate the dynamics of velocity changes in movement. For each of the above-mentioned models, whenever necessary an optimizer was considered (optimizer = "bobyqa", optCtrl = list (maxfun = 2e5)). We identified the most parsimonious models through the 'anova' function and stepwise backward elimination and choosing the model with the lowest AIC. We checked model assumptions using Q-Q plots and considered $p < 0.05$ as indicative of a significant difference between the groups.

To test whether bees performed each of the behaviours mostly inside or outside the brood area, we calculated the difference in duration between outside and inside for each behaviour for each bee. As a result, a positive value indicated that the behaviour occurred mainly outside, a negative value that it occurred mainly inside and a 0 value that the bee performed the behaviour inside and outside with equal durations. Bees that did not perform the behaviour (zero duration) were excluded from the analysis. To determine whether the duration of each behaviour was significantly different from zero, thus indicating a preference for performing the behaviour inside or outside the brood area, we conducted one-sample t-tests for each time bin.

To examine the relationships between behaviours, we employed two types of analyses: co-occurrence probabilities and time investment correlations. Co-occurrence analysis was conducted by calculating the conditional probability of a bee performing behaviour X, given that it also performed behaviour Y during the same time bin (or, for the last panel, during the previous time bin). This conditional probability $P(X_t|Y_t)$ was compared to the overall probability of the bee performing behaviour X during that bin $P(X_t)$. Behaviours were considered non-independent when the observed conditional probability significantly deviated from the overall probability, as assessed by Chi-square tests. Statistical significance was set at $p < 0.05$. For the analysis of time investment, we used Spearman's rank correlation to measure the association between the durations bees spent on two different behaviours within each time bin. This correlation was calculated for each behaviour pair within individual time bins to assess how changes in time allocation to one behaviour corresponded to changes in another. Additionally, we analysed the correlation of behaviour durations across time bins, specifically examining whether the time spent on a behaviour after a disturbance was related to the time invested in other behaviours during delayed responses. Comparisons were excluded when during the time bin considered, one behaviour was not performed at all for occurrences or performed by a single bee for durations, or when the data would be redundant (e.g., comparisons along the diagonal). Analyses and heatmaps of this section were conducted using MATLAB R2024b.

In Part II, we analysed the responses to different stimuli by first calculating the mean probabilities for each behaviour across combinations of stimulus, bin, day, colony, and test. Since mechanical disturbance was the common stimulus in both test series, a grand mean probability for "mechanical disturbance" was computed for each behaviour, bin, and colony. We then normalized the values for other stimuli using the formula log ((mean + epsilon)/(grand mean + epsilon)), where epsilon = 0.00001. The normalized probabilities were analysed using a linear model with stimulus and bin as interaction terms.

## Results

### Part I: Defensive behaviour in response to a mechanical disturbance in a whole functioning colony

**Activity patterns over time.** We first examined the velocity (Fig 2A) and location (Fig 2B) of bees over the entire duration of our recordings (10 minutes), as well as the proportion of bees performing each scored behaviour (Fig 2C), to broadly understand the colony response to a mechanical disturbance. Our data revealed distinct temporal phases. For instance, behaviours like continuous buzzing and leg raising occurred immediately after the disturbance for a short period, while others such as abdominal pumping and perching persisted over longer durations. Additionally, we observed behaviours with a delayed peak, such as pulse buzzing and grooming. To capture these dynamics, we divided the trial time into four distinct bins: "before disturbance" (the two minutes preceding the disturbance), "acute response" (the first minute following the disturbance), "delayed response" (the next three minutes), and "recovery" (the final four minutes) (Fig 2).

Based on the timing and nature of the observed activities and behaviours, we categorized them into four distinct groups: 1) general increase in activity; 2) acute behavioural responses; 3) persistent behavioural responses; and 4) delayed responses. In the following sections, the results of each of these categories are explained in detail.

**General increase in activity.** We included in this category descriptors of the bees' activity level such as their velocity and their displacement outside of the brood area, as well as the behaviours linked to locomotion i.e. flying-climbing and patrolling.

Across all time bins, worker bees moved significantly faster than queens. Comparisons across time bins indicated that following the disturbance, the velocity of both workers and queens increased significantly, particularly during the acute response phase. Queens returned to baseline velocities quickly, after about one minute, while workers showed a more gradual return to baseline over several minutes (Fig 3A; S1 Table in S1 File). This increase in movements was associated with a higher probability of being outside the brood area during the acute response phase compared to the baseline (Fig 3B; S2 Table in S1 File). This elevated probability persisted during the delayed response period but returned to baseline during the recovery. The durations of the trips outside the brood area also increased immediately after the disturbance. However, these bout durations became shorter and indistinguishable from baseline already during the delayed response period (Fig 3C; S3 Table in S1 File), denoting a gradual return to a calm state. Overall, the disturbance led to an increase in the bees' movement speed and in the time spent outside the brood area, indicating a shift from their initial brood care activity to a more excited state.

Similarly, the probabilities of bees flying-climbing or patrolling increased significantly in the acute response period. These elevated occurrences persisted during the delayed response for both behaviours but returned to baseline level during the recovery phase (Fig 4A and 4D; S4 Table in S1 File). Most workers (95%) and sometimes the queen (3 out of 8 trials) patrolled in response to the disturbance, while only a third of workers (34%) and none of the queens (0 out of 8 trials) took flight. Compared to baseline, flying-climbing bouts were longer during the acute response, and remained longer during the delayed response and recovery periods (Fig 4B; S5 Table in S1 File). Bees never patrolled before the disturbance; hence bout duration cannot be compared to a baseline for this behaviour. The duration of the bouts became gradually shorter as time passed, with bees performing shorter bouts during the delayed response and the recovery period than during the acute response (Fig 4E; S5 Table in S1 File). In terms of location, flying-climbing was performed mainly outside the brood area during both the acute and delayed responses. Patrolling, however, was observed mainly inside the brood area immediately after the disturbance, before spanning the whole arena (Fig 4C and 4F; S6 Table in S1 File).

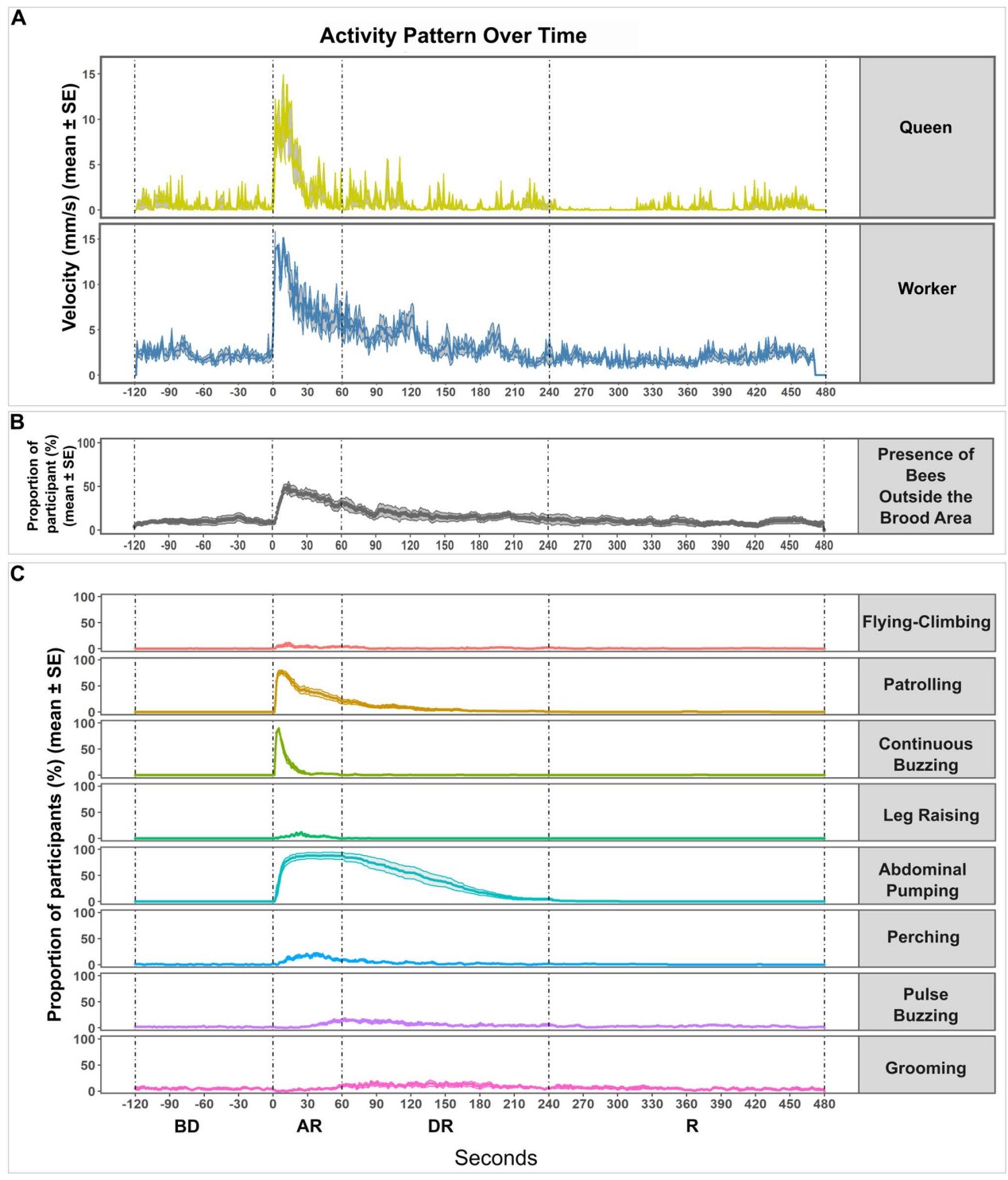

**Fig 2. Overview of the behavioural responses over time. A)** The velocity (mm/s, mean±SE) of the queen (top) and worker (bottom) over time (10 minutes). **B)** The proportion of bees outside the brood area (%) (mean±SE); **C)** The proportion of participating bees (%) (mean±SE) for eight specific behaviours. The timeline is divided into four phases (BD: before disturbance; AR: acute responses; DR: delayed responses; R: recovery) based on activity patterns (dashed lines), aligned to the disturbance (time point 0). Data from 4 colonies consisting of 20 workers with a queen, tested twice, resulting in 157 behavioural sequences (a few bees were hiding so they could not be observed).

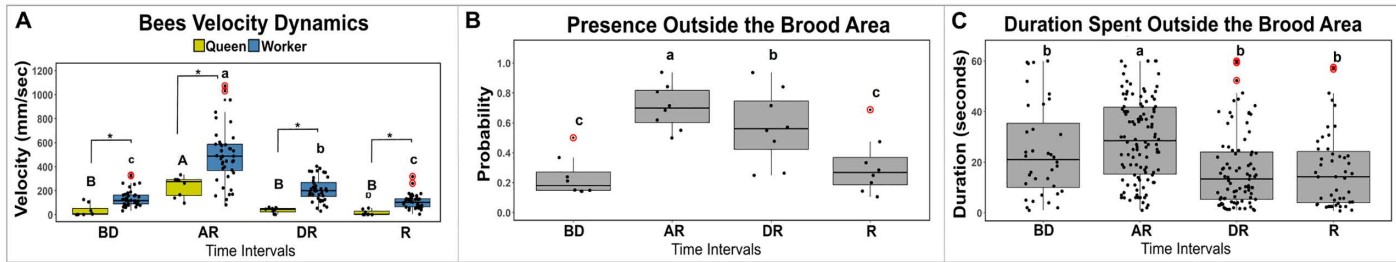

**Fig 3. Activity levels increase after the disturbance. A)** Changes in queen and worker velocity (mm/seconds) over time (BD: before disturbance; AR: acute responses; DR: delayed responses; R: recovery), in response to the mechanical disturbance. **B)** The probability of being outside the brood area across the same time bins. **C)** The duration spent outside the brood area during the same time bins. Statistical significance (P<0.01) is represented by distinct letters (uppercase for queens and lowercase for workers in panel **A**). To compare queens and workers in each time bin, an asterisk is used.

Together, these results sketch a consistent pattern: locomotor activity and exploratory behaviours peaked during the first minute after the mechanical disturbance before slowly subsiding over the course of the following three minutes. This agitated state concerned most bees and led them to move out of the brood area, spreading around their nest into the whole arena as if looking for the source of the disturbance.

**Acute behavioural responses.** Immediately after the disturbance, bees also responded with more specific behaviours such as buzzing continuously and raising their legs. The probabilities of occurrence of both behaviours increased dramatically during the acute response phase (Fig 4G and 4J; S4 Table in S1 File). Continuous buzzing was performed by most workers (94%) and all queens (8 out of 8 trials), while leg raising was only performed by a subset of the worker population (31%) and once by a queen (1 out of 8 trials). These behaviours were short-lived, as both were back to baseline occurrence probabilities already during the delayed response period. The few bouts of continuous buzzing that were performed during this period were shorter than during the acute response (Fig 4H; S5 Table in S1 File). Thus, continuous buzzing and leg raising seem to be among the most transient reactions in the defensive repertoire of bumblebees, occurring only during the peak of the defensive response. Finally, it is interesting to note that bees performed continuous buzzing preferentially inside the brood area during both the acute response and delayed response (Fig 4I; S6 Table in S1 File), whereas leg raising occurred mainly outside the brood area (Fig 4L; S6 Table in S1 File).

**Persistent behavioural responses.** The third category of behaviours that we observed includes abdominal pumping and perching. These behaviours occurred for a long time after the disturbance, with abdominal pumping continuing into the recovery phase and perching until the delayed response period (Fig 4M and 4P; S4 Table in S1 File). Abdominal pumping was not observed before the disturbance but most bees exhibited this reaction after the disturbance (92% of workers and 7 out of 8 trials for queens). Perching was also frequent (performed by 65% of workers and 7 out of 8 trials for queens). Bees started abdominal pumping immediately after the disturbance and continued throughout the acute response, with most individuals stopping during the delayed response period, indicating a gradual return to baseline activity (Fig 4N; S5 Table in S1 File). Perching bouts were longer during the acute and delayed response compared to baseline but returned to their initial durations during the recovery period (Fig 4Q; S5 Table in S1 File). Both behaviours were preferentially displayed inside the brood area (Figs 4O and 4R; S6 Table in S1 File). For perching, this may be because the nest offers an elevated vantage point over the surroundings.

**Delayed responses.** We created this final category to account for two behaviours which peaked during the delayed rather than the acute response period: pulse buzzing and grooming. Pulse buzzing occurrence peaked in probability during the delayed response, reaching intermediate levels during the acute response and recovery periods (Fig 4S; S4 Table in S1 File). This behaviour was frequent in workers (67%) but never seen in queens (0 out of 8 trials). Grooming peaked in occurrence during the delayed response and extended into the recovery period. This pattern was preceded by

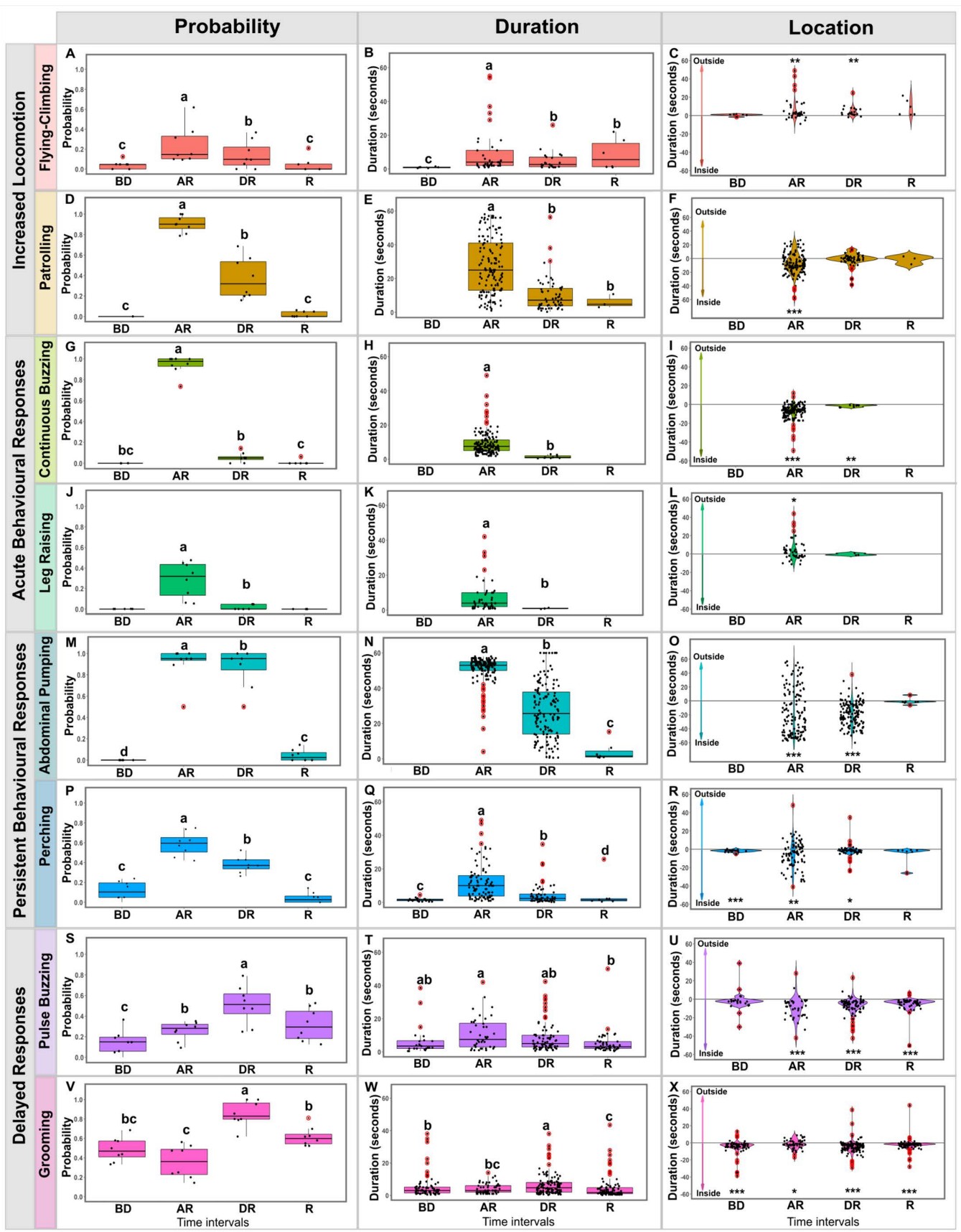

**Fig 4. Behavioural responses to a mechanical disturbance.** Probability, duration and location of eight different behaviours across time (BD: before disturbance; AR: acute responses; DR: delayed responses; R: recovery). **A, D, G, J, M, P, S and V)** The probability of occurrence of the behaviours; **B, E, H, K, N, Q, T, and W)** The duration of the behaviours normalized to one min; and **C, F, I, L, O, R, U and X)** The duration of behaviour with respect to location (inside vs. outside the brood area) during each time bin. Statistical significance (P < 0.01) is indicated by different letters. To compare the location of each behaviour in each time bin, an asterisk is used. The side on which the asterisk is displayed indicates the preferred location. Data from 4 colonies consisting of 20 workers with a queen, tested twice, resulting in 157 behavioural sequences (a few bees were hiding so they could not be observed).

a phase of inhibition, with bees being less likely to engage in this behaviour immediately after the disturbance (Fig 4V; S4 Table in S1 File). Not surprisingly, grooming was a common behaviour exhibited by both workers (95%) and queens (7 out of 8 trials). Pulse buzzing duration did not vary (Fig 4T; S5 Table in S1 File), while grooming bouts were significantly longer than baseline during the delayed response but decreased below initial levels during the recovery period (Fig 4W; S5 Table in S1 File). Both of these behaviours were preferentially performed inside the brood area (Fig 4U and 4X; S6 Table in S1 File).

**Behavioural suites displayed by individuals.** We investigated whether some behaviours were combined, or in other words whether individuals exhibited specific suites of behaviours, in three ways. First, we present the number of times that specific combinations of behaviours were performed by the same individual after the disturbance (Figs 5A-B). According to our data, a core set of four behaviours were performed by virtually all bees: patrolling, continuous buzzing, abdominal pumping and grooming. Indeed, these four behaviours were associated in 80% of individual trials, and bees performed at least three of these behaviours in 96% of the scored trials. The remaining four behaviours (flying-climbing, leg raising, perching, pulse buzzing) were less frequent and randomly associated, as can be seen by comparing our data to what independent events occurring with the same frequencies would produce (Fig 5B). Second, we determined whether performing a behaviour modified the likelihood that another behaviour would be performed by the same individual during each specific time bin (Fig 5C) or from the acute to the delayed response (Fig 5D). Third and finally, when individuals performed two behaviours within (Fig 5E) or across (Fig 5F) time bins, we examined whether the time they invested into one was correlated with the time invested into the second.

Flying-climbing was followed by grooming both before the disturbance (Fig 5C; $X^2 = 3.88$, p = 0.049) and across time bins (Fig 5D; $X^2 = 4.56$, p = 0.034). During baseline conditions, there was also a trend for pulse buzzing to trigger grooming ($X^2 = 3.55$, p = 0.059). In bees that actually performed both behaviours, there was also a positive correlation between the time invested in grooming and the time spent performing pulse buzzing (Fig 5E; $\rho(4) = 0.94$, p = 0.017).

During the acute response, bees that performed leg raising were significantly less likely to perform pulse buzzing (Fig 5C; $X^2 = 3.96$, p = 0.047) and vice versa ($X^2 = 4.21$, p = 0.040). Among the pairs of behaviours that bees performed together, the time invested in patrolling was positively correlated with the time spent buzzing continuously (Fig 5E; $\rho(133) = 0.21$, p = 0.014) but negatively correlated with the duration of leg raising and perching (leg raising: $\rho(42) = -0.34$, p = 0.024; perching: $\rho(77) = -0.31$, p = 0.005). These last observations are maybe not surprising, given that both of these behaviours require immobility while patrolling is a fast displacement. We also found a negative correlation between the durations of continuous buzzing and abdominal pumping ($\rho(129) = -0.20$, p = 0.025).

Flying-climbing during the delayed response was linked to a decreased likelihood of performing abdominal pumping (Fig 5C; $X^2 = 6.22$, p = 0.013). In addition, individuals continued doing some behaviours from the acute to the delayed response phase: this included flying-climbing (Fig 5D; $X^2 = 5.63$, p = 0.018), abdominal pumping ($X^2 = 5.56$, p = 0.018), perching ($X^2 = 8.24$, p = 0.004) and pulse buzzing ($X^2 = 13.24$, p < 0.001). Bees that maintained leg raising for a long time during the acute disturbance invested into perching afterwards (Fig 5F; $\rho(3) = 1$, p = 0.017). We also found a positive correlation between the duration of abdominal pumping during the acute response and of grooming during the delayed response ($\rho(39) = 0.48$, p = 0.001). By contrast, the bees that spent time grooming during the acute phase exhibited shorter durations of pulse buzzing during the delayed response ($\rho(4) = -0.94$, p = 0.017).

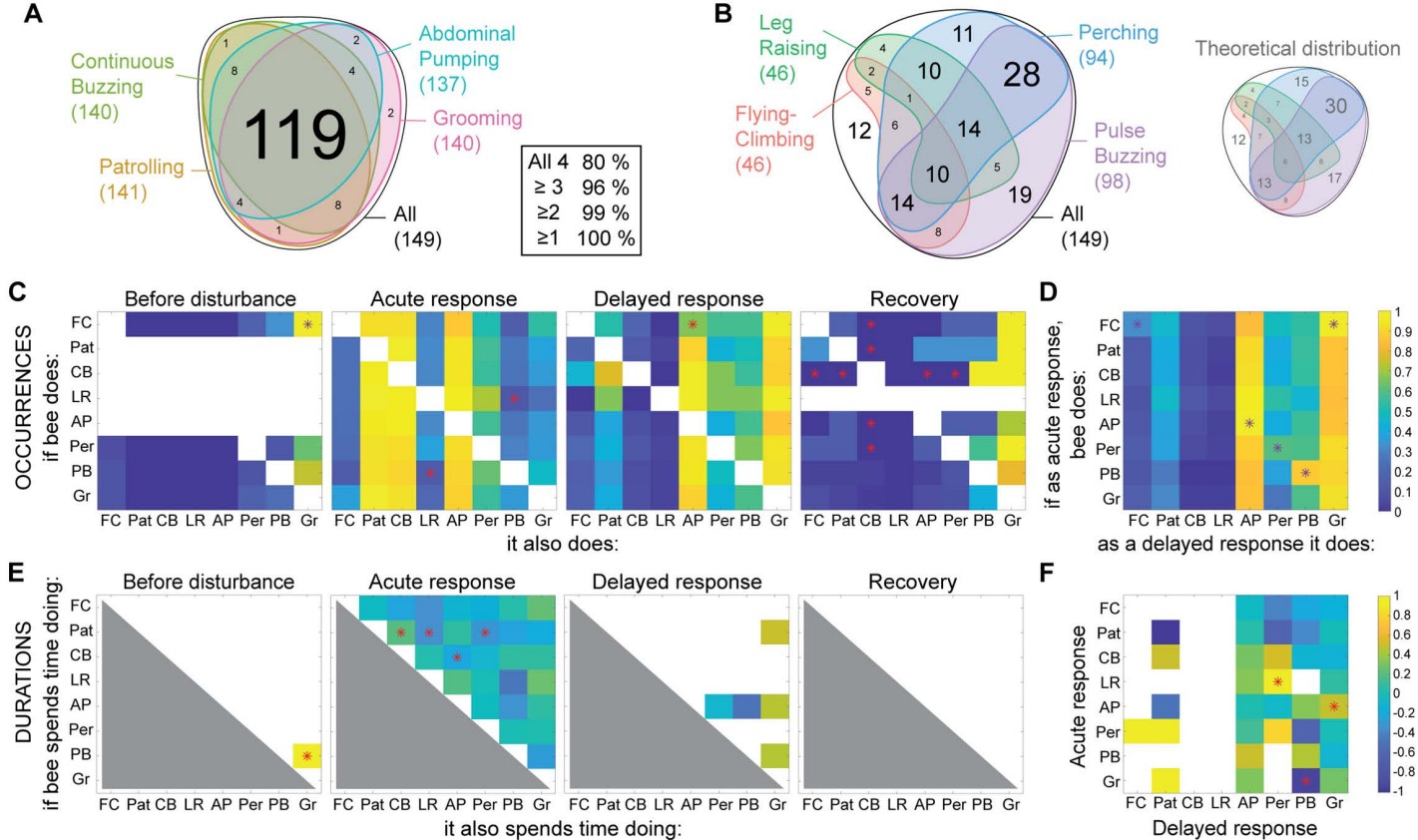

**Fig 5. Associations between behaviours exhibited by workers. A)** Venn diagram of the number of individual trials with bees performing patrolling, continuous buzzing, abdominal pumping and/or grooming. The inset indicates the percentage of trials in which the indicated number of behaviours were performed by single individuals. **B)** Venn diagram of the number of individual trials with bees performing flying-climbing, leg raising, perching and/or pulse buzzing. The smaller diagram shows the distribution expected if the behaviours were performed independently. In both **A)** and **B)**, areas with no number are empty (0 trial with this combination). **C)** Proportion of bees performing the behaviour on the x-axis, given that they performed the behaviour on the y-axis within the same bin. **D)** Proportion of bees performing a behaviour as a delayed response (x-axis) given their earlier behaviour during acute response (y-axis). **E)** Spearman's rank correlations between the durations of behaviours within each bin. **F)** Spearman's correlations between behaviour durations during acute response and delayed response. Red stars indicate significant (p<0.05) correlations, and white areas denote comparisons with insufficient data or twice the same data (diagonal). FC: flying-climbing, Pat: patrolling, CB: continuous buzzing, LR: leg raising, AP: abdominal pumping, Per: perching, PB: pulse buzzing, Gr: grooming.

Finally, bees that still performed continuous buzzing during the recovery were less likely to perform flying-climbing, patrolling, abdominal pumping, perching (Fig 5C; flying-climbing: $X^2 = 5.55$, p=0.018; patrolling: $X^2 = 11.83$, p<0.001; abdominal pumping: $X^2 = 4.65$, p=0.031; perching: $X^2 = 4.65$, p=0.031) and vice-versa (flying-climbing: $X^2 = 5.76$, p=0.016; patrolling: $X^2 = 12.00$, p<0.001; abdominal pumping: $X^2 = 4.86$, p=0.027; perching: $X^2 = 4.86$, p=0.027).

**Characterization of the different sounds produced by bumblebees.** Audio data was recorded using the same disturbance protocol but on single bees, to avoid interferences between sounds. A total of 1462 sound segments were manually annotated and assigned to one of four behavioural classes (Fig 6). Following Signal-to-Noise filtering, 840 sound segments remained (S7 Table in S1 File). To balance the sample sizes of the sound types, we randomly sub-sampled pulse buzzing segments and only used the reduced sample for the extraction of acoustic features. The random forest sound type predictions of the test set showed a match of 87% to the human classification with CI of 0.844 and 0.8989, thus confirming the validity of our manual annotations.

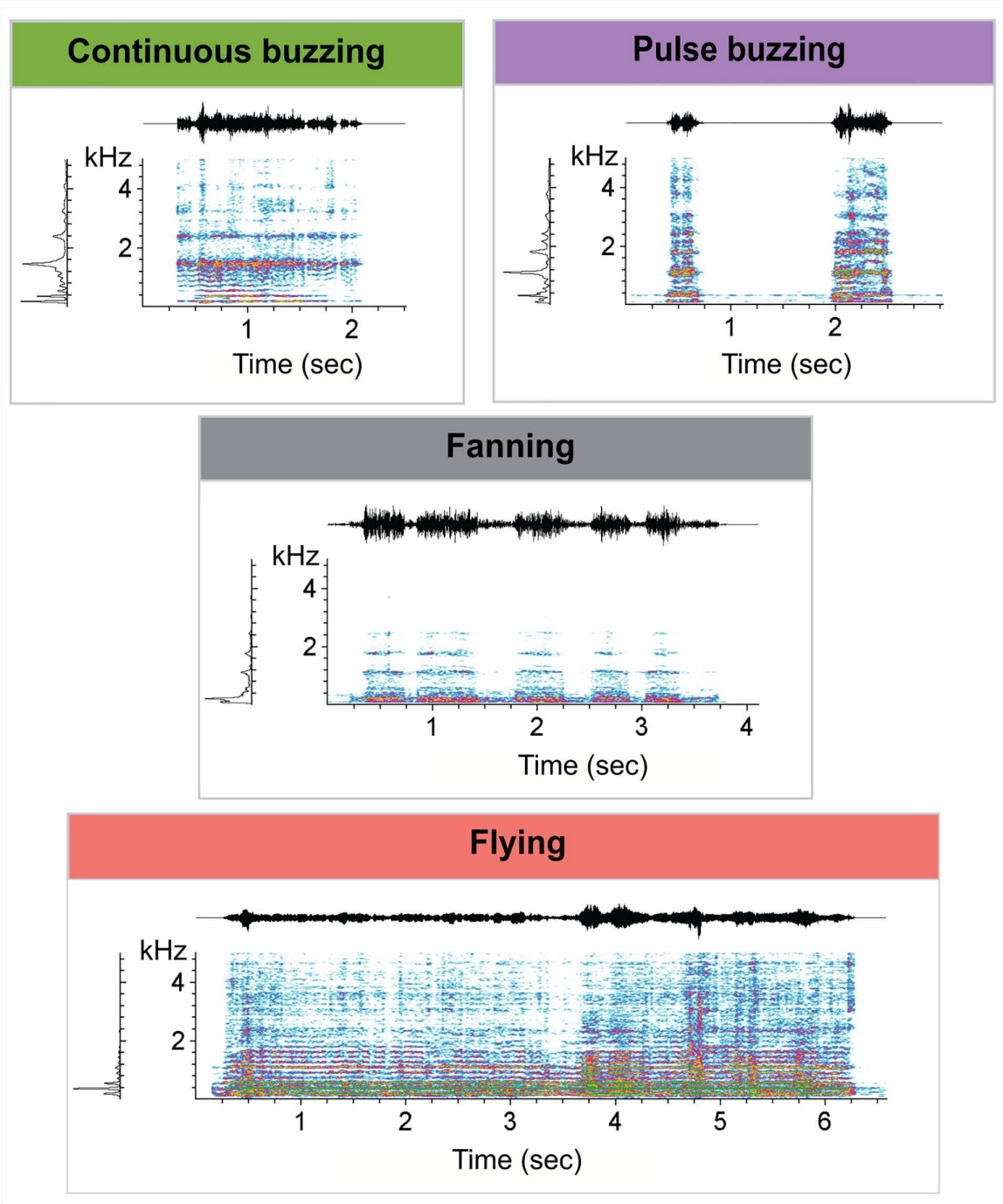

**Fig 6. A sample spectrogram of the identified and classified sound types.** Spectrograms were generated in Avisoft SASLaB Pro using 1024 FFT lengths, Hamming window and 87.5% overlap.

Analysis of the extracted acoustic features showed significant differences in the spectro-temporal profiles of the four sound types (Fig 7, S8 Table in S1 File). The fanning and flying sounds observed in the nest demonstrated lower frequency composition (lower mean, dominant and F1 frequencies, Figs 7C-E) as well as a lower entropy (Fig 7B) in comparison to pulse and continuous buzzes elicited by the disturbance. Flying produced the strongest harmonic component (highest harmonics to noise ratio, HNR) and pulse buzzing the weakest (Fig 7F). Finally, single pulses of the pulse buzzing were much shorter in duration than other buzzes (Fig 7A), although pulse series could last for much longer. Examining in more details the temporal structure of the sound sequences showed a strong isochronous pattern for pulse buzzing sequences, significantly different from null expectations (Fig 7G, S9 Table in S1 File), meaning that the sound pulses are evenly spaced and are emitted as a rhythmic sequence. The three other sound types lack any apparent temporal regularity (Figs 7H-J).

## Part II: Defensive behaviour in response to different disturbing stimuli

In Part I, we described in detail the behavioural sequence elicited by a mechanical disturbance. However, some behaviours may be specific to the type of disturbance received by the colony. To estimate this variability, we collected additional data using a set of microcolonies (consisting five workers taking care of some brood). They were tested with three different stimuli: 1) the same mechanical disturbance, 2) the introduction of a foreign object in the nest chamber and 3) air exhaled by the experimenter. Based on our previous results, we reduced the tests to three min (one min before the disturbance and two min after, corresponding to the acute response for the 1st min (AD1) and to the start of the delayed response for the 2nd min (AD2)). Each behaviour was analysed to assess how the type of stimulus influenced the probability of occurrence, taking the mechanical stimulus as a baseline to normalize the data. Perching, which is quite challenging to score, was not assessed in this dataset.

The responses elicited by the mechanical disturbance and by the breath of a potential intruder were extremely similar, with no significant differences in the occurrence of any of the behaviours scored except for presence outside the brood area (Fig 8;, S10 Table in S1 File). The bees also responded to the presence of a foreign object, as demonstrated by them leaving the brood area immediately after the disturbance and performing a number of behaviours such as abdominal pumping, continuous buzzing and leg raising (S10 Table in S1 File). The occurrence of abdominal pumping, in particular, was the same for all disturbing stimuli and across all time bins (S10 Table in S1 File). Yet, responses to the foreign object were markedly different from the other stimuli. First, we did not observe an increase in locomotion (flying-climbing or patrolling) after the disturbance (S10 Table in S1 File). As a consequence, the occurrences of flying-climbing and patrolling in the first minute after introduction of the foreign object were reduced compared to the mechanical disturbance (S10 Table in S1 File). Second, although introducing the foreign object triggered continuous buzzing and leg raising, this increase in occurrence was less pronounced than for the other stimuli: continuous buzzing during AD1 was reduced compared to mechanical disturbance, while leg raising was reduced both during AD1 and AD2 (S10 Table in S1 File). Additionally, pulse buzzing was significantly less frequent in response to the foreign object than to either the mechanical disturbance or intruder breath (S10 Table in S1 File). However, the other delayed response, grooming, occurred irrespective of the type of disturbance. Therefore, the nature of the disturbance selectively affected the behavioural responses produced. The lack of increase in locomotion in response to the foreign object, in particular, might be because bees could easily localize this potential threat, whereas the other stimuli that we used did not have a clear origin and may thus have triggered searching behaviours.

## Discussion

Defensive behaviours are essential survival mechanisms exhibited by all animal species, enabling them to protect themselves from predators [33–35] and competitors [36]. These behaviours have evolved in response to particular characteristics of threats, the context that they are encountered in and the social structure of the animals involved [34,36,37]. Common defensive responses include escape, aggressive postures, and defensive attacks [37]. In social species, these

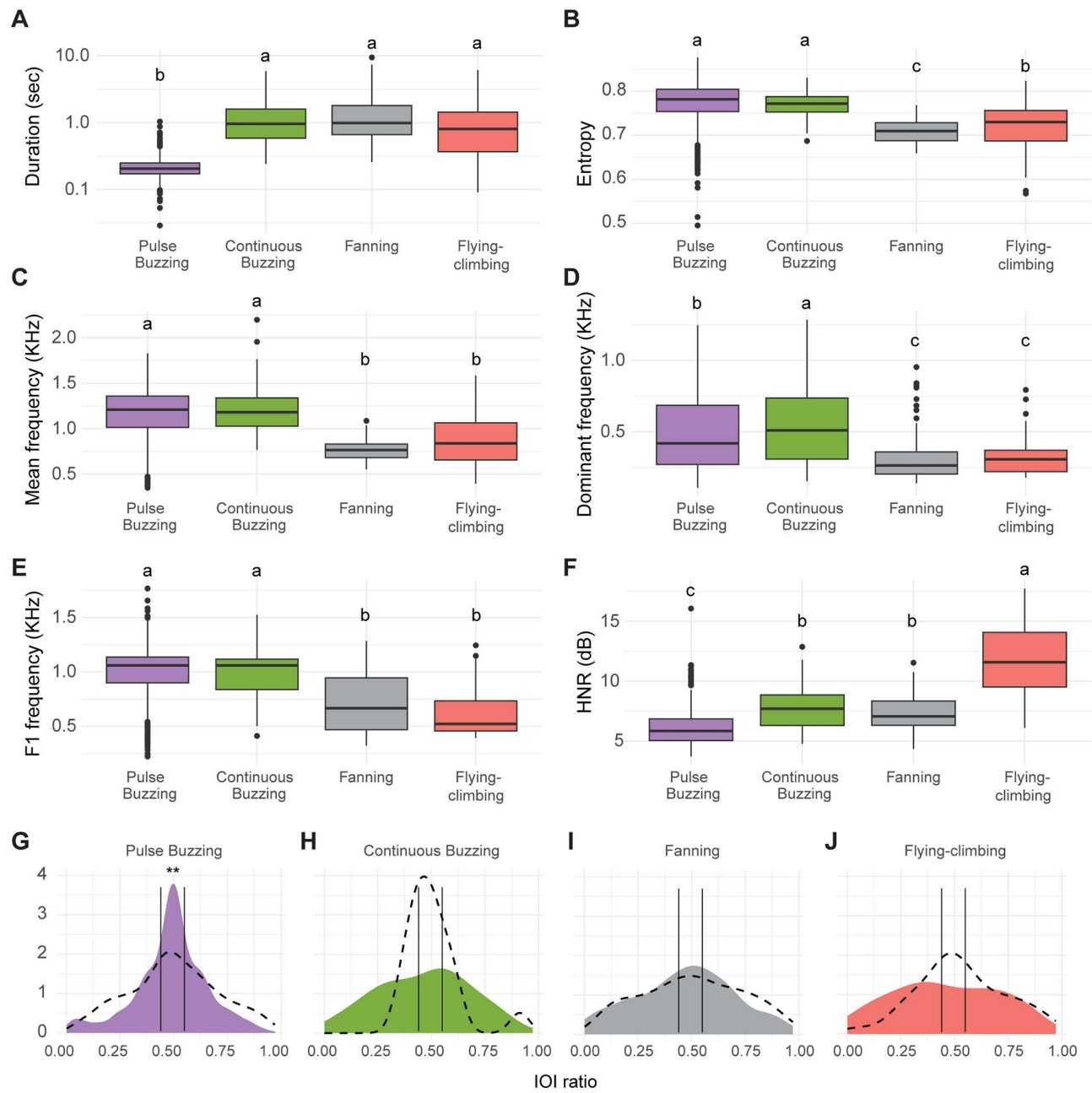

**Fig 7. Acoustic features of the four buzz types. A)** Duration – sound length (sec); **B)** Entropy – spectrographic entropy; **C)** Mean frequency – the weighted average of the frequency spectrum (kHz); **D)** Dominant frequency – an average of dominant frequency measured across the spectrogram (kHz); **E)** F1 frequency – the frequency for the second harmonic (kHz); **F)** HNR – harmonic to noise ratio (dB). **G-H)** IOI ratio distribution for symmetrical sound transition pairs. The dashed curve denotes the null ICI ratio distribution, calculated from randomly generated call times per recording while maintaining the number of annotated calls. The colored area denotes the distribution of the IOI ratios calculated from the full annotated dataset. Vertical lines mark the boundaries of isochronous IOI ratios, corresponding to 0.44 and 0.55 values. Statistical significance (adjusted for multiple comparisons with the Holm method) is indicated by different letters in panels A-F, and by stars in panel G-J.

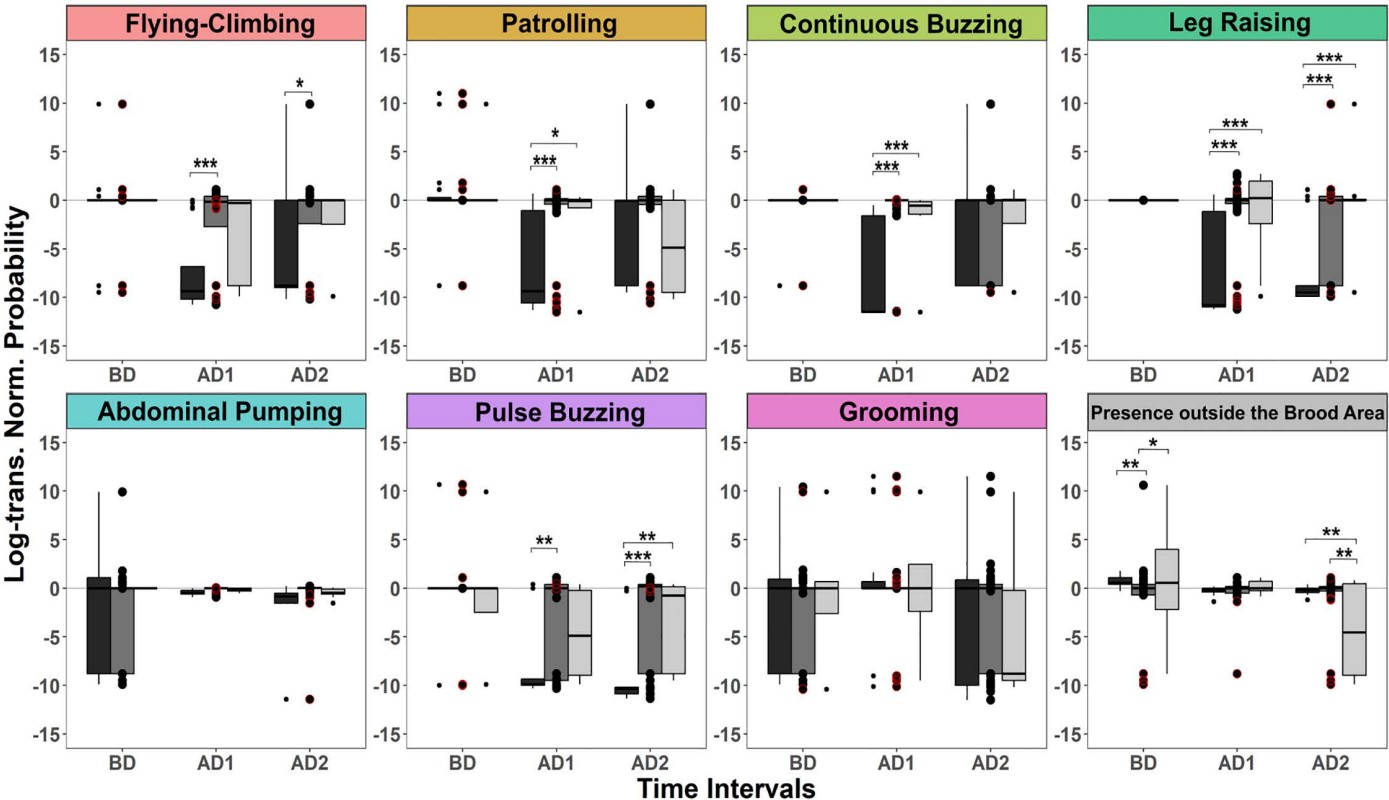

**Fig 8. Log-transformed normalized probabilities of bumblebee's behaviour in response to three disturbing stimuli.** Statistical significance is indicated by asterisk. Data from a total of 12 micro-colonies consisting of 5 workers (see Fig 1B).

behaviours can be coordinated between group members to enhance the success of the defensive response, by increasing the survival rate of defenders but also the protection of the group and its resources. This can be seen in vertebrates such as rodents [38], meerkats [39,40] and primates [41] as well as in invertebrates like honeybees [42–44] and ants [45–47], in particular through the use of warning signals such as alarm calls or pheromones [48,49]. Our study extends this understanding by providing a detailed breakdown of the defensive behaviour of a 'primitively eusocial' species, the bumblebee (*B. terrestris*). We provide a description of the full sequence of behaviours elicited by a disturbance, at both the individual and the colony level, from immediate acute responses to a return to regular activities.

Similar to previous reports in paper wasps, Polybia wasps, honeybees and other species of bumblebees [14,50–55], our results indicate that disturbing the nest triggered an immediate increase in bees' locomotion and activity levels. In contrast to queens that showed an acute increase in their activity and quick recovery, workers displayed prolonged and heightened activation and a more gradual return to baseline levels. This reiterates the task differentiation within the colony, with workers responsible for colony defence and the queens for colony reproduction rather than active defence [36,56–58]. More precisely, this increase in locomotion was caused by workers patrolling or flying around the nest, likely to search for the source of the perceived threat [15]. This interpretation is supported by these behaviours being absent when

the source of the disturbance is visible (in the case of the disturbance with a foreign object) despite other signs of agitation. Among the acute responses to the disturbance, a number of bees raised the middle and sometimes also the hind leg on one side of their body. This behaviour, referred to as the "disturbance leg-lift response", is recognized as a precursor to stinging and is therefore likely an intense and honest warning signal. The response is graded, as the number of legs lifted corresponds to the intensity of the stimulus [19]. During our tests, three bees rolled on their back completely and laid there with their stingers pointing upwards, clearly ready to sting. Given the rarity of these observations we pooled them with leg raising, considering that it may be the most extreme form of this behaviour. Acute responses may culminate into stinging or biting if the threat persists [19,59], however these behaviours are rarely described in the literature because most studies (including ours) use mechanical disturbances or $CO_2$, so that the bees do not have a physical target to attack. It would be interesting, in future studies, to include a dummy target to evaluate also those behaviours. One exception is the study by Visscher and Vetter [16] which demonstrated that after a disturbance, *B. sonorus* workers would sting a black ping-pong ball placed in front of the colony entrance but not a white one. This discrimination could also explain why the white filter paper that we used as a foreign object was not attacked. Another factor to consider in our experimental design is that our colonies were exposed to light during the disturbances, whereas they were maintained in the dark outside of experimental sessions. This may have already placed the colony in a state of alertness, since light exposure would be expected only if the nest is being dug-out by a predator. Follow-up experiments under different light regimes would be necessary to evaluate the role of this factor.

We also observed behaviours such as abdominal pumping and perching which were maintained for several minutes after the disturbance occurred. Abdominal pumping probably serves to enhance respiration, facilitating greater oxygen uptake as the insect responds to stressors with energetically demanding fight or flight behaviours [6,60–62]. Indeed, increased respiration is a common response in many species when faced with predators [63]. Virtually all bees exhibited this behaviour even when the disturbance was of low intensity, like when we introduced the foreign object, suggesting that this is a preparatory behaviour ensuring that the bee can quickly react if the threat persists or increases. Perching, where bees remain stationary but scan the environment with their antennae, also occurred within the same time window. Together, these behaviours reveal that bees maintain a state of heightened vigilance long after the disturbance happened, which likely allows them to react faster if the threat remains or returns, thus increasing the efficacy of defence.

Interestingly, the transition back to normal could not be summarized by a simple decay of all defensive responses but rather was marked by the occurrence of two behaviours: grooming and pulse buzzing. Grooming is a common behaviour that is not specific to the defensive context. Yet, more bees engaged in this behaviour a few minutes after the disturbance, for longer bouts. This may represent a recovery or stress-mitigation behaviour in bees [64]. Across insect taxa, grooming is modulated by diverse sensory inputs and neural circuits, serving not only hygienic but also adaptive functions in response to mechanical, chemical, and even visual stimuli [65]. In other animals, such as rodents, grooming is a complex behaviour that serves as an important indicator of physiological and emotional states, particularly in response to stress [66]. Alternatively, bumblebees may groom to eliminate contaminants accumulated while moving rapidly around the nest (such as dust, pollen, sugar water or wax). This hygienic behaviour could also help them get rid of potential parasites or chemicals that an intruder could have brought into the nest. Similar grooming behaviours have been reported across insects for maintaining sensory function, for example, cockroaches groom their antennae to improve their sense of olfaction by removing accumulated contaminants [67].

Consistent with previous studies, we found that bumblebees emit at least two types of buzzes after a disturbance [15,21], which we termed continuous and pulse buzzing based on their striking difference in temporal patterns. Unfortunately, these sounds have already received several names and more or less precise descriptions, making it difficult to identify them across studies. According to Kirchner and Röschard, "hissing" is a response to strong disturbances which has a fundamental frequency of 193 Hz, performed with the wings held parallel to the body axis. "Buzzing" bees, on the other hand, have their wings spread widely. Buzzing occurred after a slight disturbance, or after hissing when the colony

calms down. It is described as a low-frequency sound, while hissing significantly extends to the ultrasonic range [15]. Schneider, in contrast, observed that strong disturbances elicit a high-frequency "modulated alarm tone" produced by thorax vibrations, with the wings not contributing and held slightly elevated, parallel to the bee's body axis. Although the frequency and posture are similar to Kirchner and Röschard's hissing, Schneider also noted that this response is persistent and consists of pulses lasting 50–120 ms, features that were not reported for hissing. The same author also described the "wing alarm tone", a low-frequency response in which the bee beats her widely spread wings in response to vibrations and/or light, often while running towards the source of the disturbance [21]. This seems similar to Kirchner and Röschard's buzzing, but for the fact that Schneider places this response as a precursor to the modulated tone rather than the other way around. Our work is in better agreement with Schneider's early observations, since both the sequence and the description of his modulated and wing alarm tones match our pulse and continuous buzzing's, respectively. Some authors did not discriminate different buzz types and simply labelled this behaviour as "buzzing" [17], "fanning" [14] or "defensive buzzing" [68–70]. By providing a detailed characterization including spectrograms of the buzzes we recorded, we hope that our study will set the stage for an easier identification and more uniform naming of these sounds in the future.

Beyond this nomenclature issue, the question of the function of these buzzes is more interesting. Bees displayed continuous buzzing immediately after the disturbance and mostly while situated on top of the wax nest. Vibrations could potentially propagate to other colony members via this substrate, hence warning them of the threat. Carpenter ants, for example, use substrate-borne vibrations as alarm signals, striking their mandibles against wood to alert nestmates. Depending on intensity, ants may freeze to reduce visibility or run quickly to escape. In combination with alarm pheromones, vibrations further modulate defensive responses, potentially minimizing predation risk [71–73]. In bumblebees, this behaviour could also be directed at the intruder, scaring it through vibration and sound. While this second hypothesis already received some support [15,17,74], playback experiments will be necessary to determine if this behaviour is also used for alarm communication among the bees themselves. The function of the pulse buzzing, which peaks with a delay, is even more open to interpretations. It could be a calming signal or part of a stress alleviating response [75], so that bees regulate their physiological state, allowing them to resume faster their typical tasks such as brood care and foraging. On the contrary, it could contribute to maintaining a memory of the threat, prolonging the vigilance of the colony. Again, playback experiments will be necessary to elucidate if and how this behaviour is used in communication within the colony. The correlated performance of grooming and pulse buzzing by the same individuals seem to favour the first hypothesis, which would be intriguing given that inhibitory or stop signals are relatively rare [76].

Sociality can enable division of labour that goes beyond reproductive functions, whereby different workers perform different tasks. This differential task allocation is common also during defence, as can be seen in ants, termites, and honeybees [36,77–79]. Bumblebee colonies also seem to have workers focussed on defence in the form of guards standing at the nest entrance [5,7,8]. Yet, our analysis of the behavioural responses exhibited by individual bees revealed little evidence for a strong division of labour during the response to a vertebrate-like disturbance. Indeed, we did not find that subsets of bees performed specific suites of behaviours, which we could have expected if some workers were in charge of defending. Rather, all bees responded with a core set of four behaviours (abdominal pumping, patrolling, continuous buzzing and grooming) while other behaviours were randomly associated. One interesting exception to this pattern is that bees that maintained leg raising for a long time also invested more time into perching afterwards. This may indicate that these bees were particularly sensitive to the disturbance.

Together, our findings provide a comprehensive overview and detailed description of the defensive behaviours exhibited by *B. terrestris* and provides a solid basis for further investigations to better understand how individual and collective responses are regulated, from the (neuro)physiological underpinnings of defensive behaviours to the complex interactions between individuals and their environment that generate group-level phenotypes.

## Supporting information

**S1 File. Statistical information.** This document contains the tables S1 to S10 and their captions.
(PDF)

## Acknowledgments

We would like to thank the neurobiology group at the University of Konstanz for their helpful remarks and comments which helped us improve this work, as well as the workshop for building custom made housing arenas.

## Author contributions

**Conceptualization:** Sajedeh Sarlak, Divya Ramesh, Anja Weidenmüller, Christoph Kleineidam, Morgane Nouvian.

**Formal analysis:** Sajedeh Sarlak, Divya Ramesh, Vlad Demartsev, Morgane Nouvian.

**Funding acquisition:** Sajedeh Sarlak, Morgane Nouvian.

**Investigation:** Sajedeh Sarlak, Alica Schwarz, Lena Seitz.

**Methodology:** Sajedeh Sarlak, Anja Weidenmüller, Christoph Kleineidam, Morgane Nouvian.

**Software:** Sajedeh Sarlak, Divya Ramesh, Vlad Demartsev.

**Supervision:** Ahmad Ashouri, Seyed Hossein Goldansaz, Anja Weidenmüller, Christoph Kleineidam, Morgane Nouvian.

**Visualization:** Sajedeh Sarlak, Vlad Demartsev, Morgane Nouvian.

**Writing – original draft:** Sajedeh Sarlak, Vlad Demartsev, Morgane Nouvian.

**Writing – review & editing:** Sajedeh Sarlak, Divya Ramesh, Ahmad Ashouri, Seyed Hossein Goldansaz, Alica Schwarz, Lena Seitz, Anja Weidenmüller, Vlad Demartsev, Christoph Kleineidam, Morgane Nouvian.

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
