## [Decision Letter · Decision Letter 0]

27 Aug 2025

Dear Dr. Nouvian,

Thank you for submitting your manuscript to PLOS ONE. After careful consideration, we feel that it has merit but does not fully meet PLOS ONE’s publication criteria as it currently stands. Therefore, we invite you to submit a revised version of the manuscript that addresses the points raised during the review process.

We look forward to receiving your revised manuscript.

Kind regards,

Volker Nehring

Academic Editor

PLOS ONE

Journal Requirements:

“This research was supported by the Zukunftskolleg at the University of Konstanz and by a grant from the University of Tehran Research Vice-Chancellor, Ministry of Science, Research and Technology, and Iranian National Science Foundation (project number 97011974).”

Additional Editor Comments:

The reviewers were quite impressed with the manuscript, and I am too. One of the reviewers has a number of helpful detailed comments that you might want to consider. The quality of the figures seems fine in the full version, but perhaps you could check for consistency in capitalization (e.g. Fig. 2) and font (e.g. Fig. 6)  as suggested by the reviewer.

Reviewers' comments:

Reviewer's Responses to Questions

**Comments to the Author**

1. Is the manuscript technically sound, and do the data support the conclusions?

Reviewer #1: Yes

Reviewer #2: Yes

2. Has the statistical analysis been performed appropriately and rigorously?

Reviewer #1: Yes

Reviewer #2: Yes

3. Have the authors made all data underlying the findings in their manuscript fully available?

Reviewer #1: Yes

Reviewer #2: Yes

4. Is the manuscript presented in an intelligible fashion and written in standard English?

Reviewer #1: Yes

Reviewer #2: Yes

Reviewer #1: Thank you for giving me the opportunity to review this manuscript, “Colony defence in bumblebees (Bombus terrestris).”

In this paper, the authors characterise the behavioural response of bumblebee workers to the disturbance of their nest, analysing bout video and audio data. The ethogram created by the authors was detailed, and the methods of disturbance were well-selected: they should be commended for these, as well as their thorough analysis of the behaviours that they recorded. The discussion of these behaviours and their implications is also strong. Overall, I think the paper is good, and would recommend it for publication should my comments (and those of other reviewers) be addressed. I would like to highlight in particular that there are issues with the figures, at least in the document that I downloaded, in terms of resolution, quality and consistency: this does need to be corrected.

MAIN COMMENTS

1. This manuscript would benefit from a careful read-through for grammar and syntax issues: commas in particular are frequently in the wrong place in sentences and that means things don’t quite read smoothly.

2. I would consider pooling all information about the bees for all parts of the experiment into one section of the methods, e.g. we hear first about the four larger colonies (and also the single worker set ups) and then much later on about the microcolonies. There also isn’t any information as to how the microcolonies were set up and established, as there was for the larger colonies, so it would be better to keep this information all together. Then, just reiterate when you get to each different experiment that you used the larger colonies/microcolonies/single worker set-ups.

3. I don’t know whether it’s the system compressing the figure images oddly, but they seem very low resolution and grainy in the file that I downloaded (apart from Fig. 5 which for some reason alone is nice and crisp). In some places the text is actually impossible to read (especially in Fig. 3 and 4). From what I can tell they are probably good figures, and that’s nice to see, but the resolution issues absolutely need to be fixed. There also doesn’t seem to be much consistency in terms of formatting or fonts across figures, which is confusing.

MINOR COMMENTS

Line 20: “specific associations of behaviours: - what does this mean? I’m unclear. Do you mean whether individual bees are associated with a particular behaviour, or that they produce idiosyncratic behaviours?

Line 21: Again, “associated” is not right here. “Performed” might be better, or “produced”?

Line 25: Would “potentially signalling the threat to other colony members” fall under “preparing members for a response”? As written, it feels vague, so I would perhaps remove this.

Line 32: This is probably not how I would go about describing an animal to open a scientific paper, as it’s a little casual. I would rewrite the whole sentence to something like this: “The relatively docile temperament of bumblebees, coupled with their colourful fur, belies their capacity to defend their nests fiercely from danger. As their venom…”

Line 39: Or cuckoo bumblebees?

Line 41: I would avoid using nectar robbing here as this term is most frequently used for a specific behaviour where bumblebees bite through the corolla of a flower to steal nectar. I would term this as “nectar theft” or “resource theft” to avoid confusion.

Line 46: This might read better as: “Bombus terrestris guard

bees are surprisingly permissive to sterile non-nestmates, most often merely antennating them for longer. They also display self-grooming upon encountering them.”

(Do you mean that the guard bees groom themselves when they encounter the non nestmate?)

Line 54: Should this reference be written in the same style as the others?

Line 72: ”and very obvious to anyone who ever bothered them” - again, this reads a little too casually, I would remove or modify by swapping “bothered” to “disturbed”.

Line 82: Again, “produce specific associations of behaviours”sounds wrong: “…if individual bees are associated with specific behaviours” would be better. Unless you mean to talk about behavioural syndromes, here?

Line 92: Between which months/years?

Line 149: How much time?

Line 153: “For collecting the audio imprint of bees” is somewhat unusual wording, “To collect audio from the bees” would be better.

Line 156: The methods section is quite long, so I think you should just once again clarify here that these were the single-worker set-ups that you established (at first, I wondered how you managed to seperate out the noise from one bee from the whole group, and had to go back to realise these were individual worker set-ups!)

Line 392: if the distinction is that these behaviours occurred mostly during the acute phase, I would maybe just call this “Acute behavioural responses”.

Line 406: Again, I would perhaps terms these “persistent behavioural responses” or something - saying they are related to a “sustained alerted state” might have some assumptions in it? It also contrasts better to “acute behavioural responses”.

Line 403: I don’t quite follow the logic here. Why would continous buzzing being performed in the brood area and leg raising being performed outside of it mean that these are transient reactions? This sentence should be put before the descriptions for the locations for these behaviours.

Line 419: “Delayed responses” might be clearer.

Line 598: “digged-out” -> “dug out” (but good discussion of this limitation).

Reviewer #2: I really enjoyed reading this manuscript. It is a well-designed and elegant study that provides the first quantitative analysis of holistic defensive behaviour in this important model system. Everything was clearly described, and I couldn't identify anywhere where the manuscript was confusing or unclear. The results will be of broad interest to bumblebee biologists, as well as those interested in defensive behaviour in social insects, I congratulate the authors on a really nice piece of work

**Do you want your identity to be public for this peer review?** For information about this choice, including consent withdrawal, please see our Privacy Policy

Reviewer #1: No

Reviewer #2: No

---

## [Author Response · Author response to Decision Letter 1]

3 Oct 2025

Please see the response to reviewer file attached.

---

## [Editor Report · Decision Letter 1]

8 Oct 2025

Colony defence in bumblebees (Bombus terrestris)

PONE-D-25-35648R1

Dear Dr. Nouvian,

We’re pleased to inform you that your manuscript has been judged scientifically suitable for publication and will be formally accepted for publication once it meets all outstanding technical requirements.

This looks all good to me, congratulations on this nice piece of work! Please do make the dryad submission available, the doi doesn't come through yet.

Kind regards,

Volker Nehring

Academic Editor

PLOS ONE
---

## [Editor Report · Acceptance letter]

PONE-D-25-35648R1

PLOS ONE

Dear Dr. Nouvian,

I'm pleased to inform you that your manuscript has been deemed suitable for publication in PLOS ONE. Congratulations! Your manuscript is now being handed over to our production team.

Kind regards,

on behalf of

Dr. Volker Nehring

Academic Editor

PLOS ONE